# Exploring the Role of Vitamin D, Vitamin D-Dependent Proteins, and Vitamin D Receptor Gene Variation in Lung Cancer Risk

**DOI:** 10.3390/ijms25126664

**Published:** 2024-06-17

**Authors:** Tudor Ciocarlie, Alexandru Cătălin Motofelea, Nadica Motofelea, Alina Gabriela Dutu, Alexandra Crăciun, Dan Costachescu, Ciprian Ioan Roi, Ciprian Nicolae Silaghi, Andreea Crintea

**Affiliations:** 1Department VII Internal Medicine II, Discipline of Cardiology, University of Medicine and Pharmacy “Victor Babes”, 300041 Timisoara, Romania; ciocarlie.tudor@umft.ro; 2Department of Internal Medicine, University of Medicine and Pharmacy “Victor Babes”, 300041 Timisoara, Romania; 3Department of Obstetrics and Gynecology, University of Medicine and Pharmacy “Victor Babes”, Eftimie Murgu Sq. No. 2, 300041 Timisoara, Romania; nadica.motofelea@umft.ro; 4Department of Molecular Sciences, Iuliu Hațieganu University of Medicine and Pharmacy, 400349 Cluj-Napoca, Romania; alina.dutu@umfcluj.ro (A.G.D.); acraciun@umfcluj.ro (A.C.); silaghi.ciprian@umfcluj.ro (C.N.S.); crintea.andreea@umfcluj.ro (A.C.); 5Radiology Department, University of Medicine and Pharmacy “Victor Babes”, 300041 Timisoara, Romania; costachescu.dan@umft.ro; 6Multidisciplinary Center for Research, Evaluation, Diagnosis and Therapies in Oral Medicine, University of Medicine and Pharmacy “Victor Babes”, Eftimie Murgu Sq. No. 2, 300041 Timisoara, Romania; ciprian.roi@umft.ro

**Keywords:** lung cancer, vitamin D, vitamin D receptor, single nucleotide polymorphisms

## Abstract

Lung cancer has an unfavorable prognosis with a rate of low overall survival, caused by the difficulty of diagnosis in the early stages and resistance to therapy. In recent years, there have been new therapies that use specific molecular targets and are effective in increasing the survival chances of advanced cancer. Therefore, it is necessary to find more specific biomarkers that can identify early changes in carcinogenesis and allow the earliest possible treatment. Vitamin D (VD) plays an important role in immunity and carcinogenesis. Furthermore, the vitamin D receptor (VDR) regulates the expression of various genes involved in the physiological functions of the human organism. The genes encoding the VDR are extremely polymorphic and vary greatly between human populations. To date, there are significant associations between VDR polymorphism and several types of cancer, but the data on the involvement of VDR polymorphism in lung cancer are still conflicting. Therefore, in this review, our aim was to investigate the relationship between VDR single-nucleotide polymorphisms in humans and the degree of risk for developing lung cancer. The studies showcased different gene polymorphisms to be associated with an increased risk of lung cancer: *TaqI*, *ApaI*, *BsmI*, *FokI*, and *Cdx2*. In addition, there is a strong positive correlation between VD deficiency and lung cancer development. Still, due to a lack of awareness, the assessment of VD status and VDR polymorphism is rarely considered for the prediction of lung cancer evolution and their clinical applicability, despite the fact that studies have shown the highest risk for lung cancer given by TaqI gene polymorphisms and that VDR polymorphisms are associated with more aggressive cancer evolution.

## 1. Introduction

According to the American Lung Association, mortality from lung cancer is higher each year than from breast, colon, and prostate cancer combined, with lung cancer remaining the major cause of cancer-associated deaths. Even though smoking rates have dropped by nearly 7%, there are still approximately 34 million smokers in the United States alone. According to the Romanian National Institute of Public Health 2023 report, Romanian men are more prone to being diagnosed with lung cancer, with pulmonary cancer accounting for 17% of the newly reported oncological cases compared to 7% in women. The same report shows that the number of male smokers is 4 times higher than female smokers (30.6% compared to 7.6%) [1]. Corroborating these numbers with the dramatic proportion of 90% of lung cancers caused by this vice, more than 30 million cases of pulmonary neoplasm are expected to arise in the following years [1,2,3]. The risk factors that dramatically influence the chances of developing lung cancer are numerous and relatively well-known. Besides the above-mentioned smoking habit, a positive family history increases the likelihood of developing a lung tumor two-fold. Furthermore, with air pollution reaching alarming records annually, environmental factors constitute a significant variable to be taken into consideration. Occupational exposure to chemicals such as Radon, Asbestos, Uranium, Chromium, and others also plays an important role in the evolution and incidence of this disease [4,5,6]. Biological agents can also be held responsible for a lung tumor’s emergence, mainly represented by HPV and *Mycobacterium tuberculosis*, which may cause severe infections and, finally, could develop into lung cancer. Other marginal factors, such as dietary habits, hormonal changes, and the presence of diabetes mellitus, were also reported [4,7].

However, if diagnosed early, more precisely during the first stage, this serious condition can reach a 5-year survival rate of 62%. To support early diagnosis capabilities, scientists have developed not only novel treatments but also reliable diagnostic and prophylactic tools. Biomarkers obtained from blood, airway epithelial cells, or even breath can be used nowadays to detect pulmonary cancer at its early stages [8,9,10,11,12,13].

Adjacent to these risk factors, a relatively underestimated factor is represented by the existence of single-nucleotide polymorphisms (SNPs), which have been intensely studied in recent decades in the context of the evolution of lung cancer or other types of neoplasms [14,15,16]. An SNP describes a site in the genome that has a different DNA base in at least 1% of a population and is one of the most important factors that provide genetic variability in an individual’s genome. Knowing that a single base change can already characterize a mutation, we have to rely on DNA sequencing to understand the distinction better. A single base change that is considered a harmless SNP in one population can be influenced differently by genes or the environment and considered a mutation in another population. Thus, we have to look at frequency and gene function to understand the implications in processes like immune regulation, cell cycle mechanisms, or DNA mismatch repair, which are crucial in the context of abnormal cellular growth phenomena [17,18]. In addition, the newly improved capabilities to perform transcriptomic and proteomic studies also revealed that the expression levels of several proteins might considerably differ in normal tissues when compared to tumoral samples, a fact that suggests a possible interference of these genetic regulations in the mechanisms of oncogenesis [17,19,20].

Finally, in the last two decades, scientists investigated the hypothesis that vitamin D (VD) may be a key factor in lung cancer emergence, with studies proving the link between this micronutrient along with its analogs and lung cancer. Thus, corroborating the information about the aforementioned detrimental genetic variations with the implications of VD in the development of pulmonary tumors, research on the genetics of the main VD-interacting proteins is highly relevant and may provide factual and valuable results [21,22,23,24].

In this review, our aim was to emphasize the importance of VD-dependent proteins and the VD receptor (VDR) in lung pathology and propose a new model of research for the prognostic accuracy of the risk of developing lung cancer. A better understanding of the molecular mechanism of oncogenesis may come with new findings in the field of oncology and cancer therapy.

## 2. The Role of Vitamin D and Vitamin D Receptor in Immunity and Carcinogenesis

### 2.1. Vitamin D and Its Role in Immunity and Carcinogenesis

Widely known as a principal regulator of calcium–phosphate balance, VD is a vital nutrient for many other processes in the human body. Referred to as calciferol, this fat-soluble vitamin is provided both exogenously, from foods like egg yolks, fish, red meat, or liver, and endogenously, during exposure to UVB radiation. Significant deficiencies of this nutrient are common among certain vulnerable groups, such as breastfed infants, populations with dark skin tones, older adults, and people with different medical disorders. For instance, a 13% rate of severe calciferol insufficiency was reported among European citizens, while 40% of people are estimated to have a moderate shortage of this nutrient in the same population. With one in four Americans also suffering from average VD deficiency and 490 million people suffering from the same deficit in India, there are important global consequences regarding poor administration of this nutrient [22,25,26].

Two forms of this vitamin are worth mentioning: VD_2_ (ergocalciferol) and VD_3_ (cholecalciferol), which are mainly differentiated by their origin. Plant/fungi-sourced foods are rich solely in VD_2_, while VD_3_ is found only in animal-based food products and can be synthesized through sun exposure. VD_3_ has a higher potential of improving the circulating levels of VD [27,28,29].

The common clinical effects caused by insufficient VD are relatively well-known: reduced bone density, muscle spasms, or aches and rickets in children. However, researchers have also discovered more profound mechanisms that may be influenced by VD, from high blood pressure and diabetes development to the onset of autoimmune diseases or even cancer [30].

One of the most important roles of VD is the regulation of the immune system, having different effects on immune cells such as macrophages, monocytes, dendritic cells, T-cells, and B-cells [30]. Monocytes are involved in infection defense, as they produce inflammatory cytokines. VD induces the production of peptides such as β-defensin 2 and cathelicidin, inhibits NF-KB activation, and modulates the epigenetics of macrophages, inducing subtype differentiation and even autophagy [31]. Dendritic cells play the role of antigen-presenting cells, activating adaptive immune responses. Active VD reduces the expression of stimulatory molecules on dendritic cells and the expression of *MHC II* molecules. In addition, VD suppresses the production of dendritic cell cytokine IL-12 and IL-23 but promotes the expression of cytokine IL-10 [29,31,32]. T-cells interact closely with antigen-presenting dendritic cells. Both of them express VDR and *CYP27B1*, therefore VD also has an effect on T-cells by suppressing their proliferation and differentiation, as well as decreasing the secretion of inflammatory cytokines. In the case of B-cells, VD inhibits the differentiation into memory cells and plasma cells, decreasing B cells’ overall function [29,30,32,33,34].

The prevalence of VD deficiency is of critical importance to public health and needs to be addressed in order to improve the health outcomes of the general population by ensuring access to adequate nutrition and healthcare [35,36,37,38].

VD supplementation was linked to improved cardiovascular and metabolic health, decreased inflammation, improved blood pressure, and cancer prevention. VD promotes the expression of genes involved in apoptosis, cell cycle arrest, and differentiation, which can inhibit the development of cancer. Additionally, it reduces oxidative stress in cells, which is a major contributor to the development of numerous types of cancer [26,38,39].

We summarize the involvement of VD in the interplay between immune and cytotoxic processes in Figure 1.

Several studies have associated high circulating VD levels with a decreased incidence of lung cancer. In the research conducted by “Boughanem et al.” [40] with over 1500 participants, a significantly lower risk of lung cancer was revealed among those with higher levels of VD intake, suggesting a protective effect of VD against the development of lung cancer. The authors also found that the risk appeared to decrease further with higher VD intake. They concluded that a diet with adequate amounts of VD could potentially reduce the risk of lung cancer [41]. On the same note, according to Norton et al. [41], epidemiological studies suggest that a lower incidence of lung cancer is directly correlated with higher vitamin D levels [42]. Zhan et al. defined the daily intake of VD associated with the risk of developing NSCLC: at <4 µg/d, the risk associated with developing NSCLC is the highest, decreasing to more than half after >10 µg/d intake [43]. However, further research is needed to confirm the link between VD and lung cancer risk.

In another study conducted by “Wang et al.” [42], the authors examined the data of 4843 participants from the China Kadoorie Biobank, analyzing the associations between serum VD levels, lifestyle factors, and the risk of developing lung cancer. They found that those with a serum VD level of 30 ng/mL or higher had a 27% lower risk of developing lung cancer compared to those with a VD level below 20 ng/mL. Additionally, the authors noted a dose–response relationship, with a higher VD level leading to a greater reduction in risk. They also evaluated the potential impact of lifestyle factors such as smoking and alcohol consumption and found that VD levels still had a protective effect, even after accounting for these lifestyle factors [25]. VD reduces the risk of lung cancer by suppressing the activity of certain genes that are associated with the occurrence of the disease. VD supplementation reduces the risk of lung cancer in both smokers and those with a positive family history of lung cancer [25]. Ramnath et al. [44] observed a favorable effect, especially in never-smokers, suggesting that even in patients with a higher risk of developing lung cancer, VD supplementation could be beneficial.

With an innovative approach, McFarland et al. [25] found that VD deficiency interferes with the physiology of the anti-tumoral immune responses due to induced depression, regularly caused by the shortage of this nutrient in the systemic circulation. Thus, from a total of 98 patients with lung metastases, those with the lowest VD blood levels and associated depression were associated with the worst survival prognosis. Even though the limitation of the study was noticeable, considering the known pre-existing medical condition of the patients, which might induce depression solely by itself, the idea that nutritional guidance and psychological aid may be considered in synergy with conventional therapy schemes is not only unique but also encouraging for both patients and researchers [45]. As a genetic regulator, VD was also considered as a possible mutagenic agent, especially considering its implications in multiple signaling pathways, which include but are not limited to cell cycle regulation or the biosynthesis of oncogenes and lymphokines.

Autier et al. [44] clearly underlined the potential benefits of VD supplementation with regard to lung cancer through a cohort study of 11,721 participants. The participants who had taken VD supplements of at least 800 IU daily had a 22% lower risk of developing lung cancer compared to those who did not take the supplement. This desirable effect is more pronounced in higher doses of VD or in those participants with a higher baseline of VD [46]. Moreover, Liu et al. [45] showed in a meta-analysis that VD deficiency was significantly associated with an increased risk of lung cancer: the lower the VD levels, the higher the risk of lung cancer. Therefore, maintaining adequate levels of VD is crucial for optimal health and reducing the risk of developing cancer [47].

In addition to the direct effects of VD on the human body and lung oncological pathology, VD also indirectly influences the health status through the molecules dependent on the presence of VD, referred to as VD-dependent proteins [46,47], and also through the VDR, which will be discussed further.

These findings suggest that supplementation may be an effective way to reduce adverse reactions and improve health outcomes. Due to its ability to boost immunity and potentially protect against cancer, it is important to ensure that the population is well informed of the potential benefits of VD and encouraged to incorporate sufficient amounts of VD-rich foods into their daily diet.

### 2.2. Vitamin D-Dependent Proteins in Lung Cancer

The most widely acknowledged VD-dependent proteins are members of the calbindin family. Discovered more than 50 years ago, this family is mainly represented by three members: calbindin 1 (CABP28K), calbindin 2 (CABP29K/calretinin), and calbindin 3 (CABP9K/S100G). Calbindins are intracellular proteins that have a very high affinity for calcium ions, being assigned to the EF-hand superfamily since they contain the EF-hand structural motif [48,49].

Calbindin 1 is encoded in the human body by the *CALB1* gene, with a molecular weight of 28 kDa. Four of the six EF motifs are responsible for Ca^2+^ ions binding, without notable structure alterations between the native protein and its calcium-loaded conformation [50].

Recent evidence based on former assumptions suggests that CABP28K may also be relevant in the context of lung cancer management [51]. Firstly, at a wide level, the overexpression of *CALB1* was shown to suppress apoptosis in the tumoral cells, presumably resulting in a worse outcome of neoplastic diseases [50,51,52]. Jin et al. [53] even suggested that *CALB1* may be considered a target for lung cancer therapy since they demonstrated that downregulation of this gene via miR-454-3p mature miRNA notably reduces tumoral development in NSCLC patients.

Castro et al. [54] reported that calbindin 1 is found in 74% of the 452 lung cancer tissues. Interestingly, the authors found conflicting results regarding the correlation between the presence of this gene at the tumoral site and the survival prognosis, concluding that patients with CABP28K-positive tumoral cells had more positive chances of survival than those with *CALB1*-negative carcinomas. On the other hand, considering that former evidence suggests the involvement of calbindin 1 in the modulation of secretory mechanisms [51,53,55], its presence at the tumoral sites may determine better-differentiated adenocarcinomas, eventually helping clinicians to perform tumor resection surgeries with higher rates of success [51,55].

Conclusively, even if the limited current literature resources are still contradictory or at least cannot provide a full picture of the CABP28K implications in lung cancer, it is clear that this protein may serve as a potential primer for novel therapeutic strategies that aim to alleviate lung carcinoma.

Encoded by *CALB2* in the human body, calretinin is a calbindin with a molecular weight of 29 kDa. Being involved in the same processes as CABP28K, these two proteins slightly differ in some aspects. For instance, even if both of them are mainly expressed in the central nervous system, calretinin can be found only in colon tumors and not in normal tissue, in contrast to calbindin 1 [56]. Furthermore, they have different calcium-buffering capacities, with CABP29K having five motifs for calcium binding, more than the first one, and implicitly, they also have different structures [57].

CABP29K is considered an important marker of mesothelioma, which is a type of cancer usually triggered by Asbestos, with its origins in the mesothelium (a fine layer that coats most of the internal organs) [57]. The real interest shown for calretinin in this context is due to its occasional presence observed in regular cases of lung cancer. Thus, by corroborating this fact with data that establish CABP29K as a typical mesothelioma marker, which is an aggressively malignant cancer, the conclusion is that diagnostic and therapeutic strategies may seriously interfere with the questionable expression of calretinin in lung tumors [58]. However, this issue can be easily overcome by a simple strategy proposed by researchers: the screening of other mesothelioma-specific markers, such as podoplanin, cytokeratin, or mesothelin, which can help to differentiate lung carcinoma from mesothelioma [59]. Finally, it is worth mentioning that the significance of the calretinin-based diagnosis of lung cancer is, however, very limited since there are multiple studies showing a poor overall correlation between lung tumors and CABP29K expression, with 40% or even fewer lung cancer sites expressing this protein [59].

The final relevant member of the calbindin triad is a protein of just 9 kDa, known as protein S100-G. Having a much lower calcium buffering capacity, with just two calcium binding sites, from which one has a low affinity for these ions, CABP9K is found mainly in the duodenum and jejunal mucosa, in contrast with the other two calbindins, which are predominantly expressed in the central nervous system [52]. Even if the S100G implications in lung cancer were just relatively recently uncovered and the bibliographic resources are rather limited, there were several reports of its overexpression in lung cancer, especially in the Human Protein Atlas database and also in a study carried out by Liu et al. [60]. This protein was also overexpressed in breast and renal cancer, facts that may encourage further studies on its possible modulation of pulmonary neoplasms [61].

Last but not least, all these calbindins may be correlated with the pathophysiology of lung cancer through the inflammatory processes. This is based on the observation that calcium modulation was proven to be important in many diseases that exhibit inflammation, such as Alzheimer’s and diabetes, or in inflammation-associated bone resorption, which was correlated with calcium presence, even if the effects of an increased/decreased calcium level led to different responses for different inflammatory processes [50,52,62].

The expression of the VD-dependent proteins in lung cancer is summarized in Table 1.

### 2.3. Vitamin D Receptor in Lung Pathology

Localized at the nuclear level, the VDR is the main receptor for the bioactive form of VD_3_, acting as a regulator of the calcitriol effects on cells. There are currently three known isoforms of the VDR: VDRA (427 amino acids), VDRB1 (477 amino acids), and a truncated form of the VDRA-FokI variant (424 amino acids), with VDRA being the most frequent isotype. The polymorphic N terminus in human VDR isoforms influences transcriptional activity by modulating the interaction with transcription factor IIB [64]. A brief glimpse into the VDR working mechanism, as shown in Figure 2, shows that after VD_3_ binding, the receptor enters the nucleus, forming heterodimers with the retinoid X receptor (RXR), another type of nuclear receptor. These aggregates will finally lead to the transcription of VD_3_ target-responsive genes, upon binding to punctual transcriptional regulators on DNA [65]. This receptor greatly influences calcium homeostasis through different mechanisms. Firstly, in calciferol-binding dependence, it directly promotes the activation of gene transcription for *TRPV6*, which is a Ca^2+^ channel in the intestinal cells, thus favoring intestinal calcium absorption [66]. At the same level, after VD binding, the VD/VDR complex can determine the upregulation of claudins 2 and 12, as well as of other similar tight-junction proteins like cadherin-17 or aquaporins, in this way promoting the absorption of calcium and regulating the fluxes of this ion through the entire intestine. The tight-junction CLDN2 gene is a direct target of VDR [64,65]. Secondly, the VD/VDR aggregate has the capacity to stimulate catabolic processes in bones on a background of hypocalcemia, in order to mobilize all calcium deposits, while also inhibiting osteoblasts activity. Furthermore, this ligand–receptor complex also indirectly influences renal calcium reabsorption, in strict correlation with serum calcium fluctuations, maintaining physiological values of Ca^2+^ in the systemic circulation. According to the GeneCards database, the most common disease associated with the VDR is Vitamin D-dependent Rickets. Complementary to this, according to UniProt and its affiliated sources, VDR is involved in important cell regulatory mechanisms, such as cell differentiation, cell morphogenesis, decidualization, or cellular calcium ion homeostasis, mechanisms that, if malfunctioning, may inevitably be related (but not limited) to cancer emergence and progression [67].

Multiple studies have shown the relevance of the VDR in a great palette of cancers, ranging from prostate, skin, bladder, colon, ovary, breast, kidney, and lung to non-Hodgkin lymphoma, hepatocellular carcinoma, or thyroid carcinoma [67,68,69,70].

In the context of lung cancer, the VDR was formerly associated with better survival outcomes in lung cancer patients, while also showing the antiproliferative effect of the receptor in cell lines. It is presumed that one of the mechanisms through which the VDR inhibits tumoral growth is by maintaining a normal balance between oncogenic and cancer-suppressing lncRNAs, an observation demonstrated in studies on mice that lack the VDR and consequently exhibit a more oncogenic lncRNAs profile [71]. LncRNA profiling reveals new mechanisms for VDR protection against skin cancer formation. An evaluation of the expression of the VDR related to lncRNAs is relevant in lung cancer. In addition, normal VDR structure and functioning also indirectly result in tumoral anti-proliferative mechanisms due to the mediation of VD actions, which is largely regarded as an antitumoral agent [71,72,73].

Furthermore, a study investigating a possible correlation between VDR levels and survival outcomes in patients with lung adenocarcinoma concluded that higher levels of this receptor were associated with improved survival rates in the patients. This was explained by a reduction in S-phase transition-stimulating proteins, such as S-phase kinase-associated protein 2, Cyclin 1, or Retinoblastoma-associated protein, which may result in mediated G1 arrest and consequently in reduced proliferation of the tumoral cells [74,75,76]. The conclusions show that VDR deficiency can result in gradual lung function decline due to abnormal cellular signaling, which will not only affect the normal immune processes but will also favor inflammation and alter mechanical lung functions, facts that may finally lead to an early-onset of chronic obstructive pulmonary disease [74].

The VDR also plays an important role in the physical defense mechanisms and even in keeping the integrity of lung tissue. Since the VD/VDR complex influences the transcription of tight-junction proteins, dysregulations in the functioning of the receptor may directly result in the disruption of the pulmonary epithelial barrier. VD/VDR signaling attenuates lipopolysaccharide-induced acute lung injury by maintaining the integrity of the pulmonary epithelial barrier. This may ultimately lead to impaired gas exchange and the onset of acute lung injury or even acute respiratory distress syndrome in more severe cases of this condition. Expectedly, it was shown that mice that lack the VDR were more susceptible to lipopolysaccharide-induced injuries and developed more severe forms of lung damage than their wild-type littermates, a fact that also highlights the role of this receptor in the integrity maintenance of pulmonary tissue. Finally, Zheng et al. [77] showed that a physiological relationship between VD and the VDR may result in enhanced epithelial tissue repair after a lung injury.

## 3. Vitamin D Receptor Gene Variation and Lung Cancer Risk

Polymorphisms of the VDR are largely responsible for pulmonary neoplasm onset in a presumable ethnicity-, age-, and gender-dependent manner. Consequently, Cdx-2, Bsm1 Taq1, and Fok1, which are VDR polymorphisms, were found to be the alterations with the highest correlations with lung cancer development.

Even if Kim et al. [74] found a correlation between lung cancer and VDR a decade ago, very recent studies still militate for prudence and further investigations in order to establish clear lines when approaching anti-tumoral therapy strategies that include VDR gene regulation. Research conducted by Dogan et al. aimed to investigate the effects of VDR gene variations on lung cancer risk. The research included a total of 191 individuals with lung cancer and another 291 individuals without lung cancer. The results indicated that polymorphisms in the VDR gene are associated with a greater risk of lung cancer compared to those without such variations. Moreover, the researchers found that the risk of lung cancer increased with the number of VDR gene variations present in an individual [78]. This provides evidence that VDR gene variations may be an important factor in lung cancer risk. These results highlight the need for further research to better understand the exact role of VDR gene variations in lung cancer risk [79].

Other studies showed that not only should the VDR gene polymorphism phenomenon be considered responsible for an increased risk of lung cancer but also individuals’ ethnicity in corroboration with this genetic diversity. The main and the most frequent polymorphisms investigated for this purpose were ApaI (rs7975232), BsmI (rs1544410), and TaqI (rs731236). The results found in several research studies are summarized in Table 2, comprising the relationship between VDR genetic variants and the ethnicity of subjects, in rapport with the degree of risk for developing lung cancer [80]. Although far from being exhaustive, Table 2 is still a comprehensive and perhaps inspiring source for future research studying the impact of polymorphisms on pulmonary or other different types of cancer. In addition, it may be considered by clinicians for rapid and on-site prediction of the population incidence, individual risk factors, or even personal survival/cancer management prognostics, relying plainly on genetic information gathered from a pool of patients or an individual.

Another significant observation based on VDR gene polymorphism regards the disparities observed between different studies on the same genetic variants. For instance, while Li et al. [79] could not find any correlation between *ApaI* genotype variations and lung cancer, Kaabachi et al. [85] found a strong association between these two variables.

Furthermore, it is important to mention that these alleles may have an even larger clinical impact and significance when studied together in groups. Such an example is provided by Li et al. [90], where different variants of *ApaI*, *Cdx2*, and *FokI* are considered simultaneously in the context of lung cancer risk. The main conclusion of the study revealed that *CC-AA (Apa1-Cdx2)* and the *CC-AA-CC (ApaI-Cdx2- FokI)* haplotypes were associated with higher lung cancer incidence, taking this kind of prospective cancer development prognostic tool to a whole new level with enhanced specificity and improved accuracy.

Besides the above-mentioned polymorphisms, VDR mRNA expression is also important in lung cancer evolution. The presence of this receptor is modulated by other ligands (apart from VD), such as several polyunsaturated fatty acids, curcumin, and lithocholic acid, which, even if having a low affinity for this receptor, may play an important role as anti-tumoral agents [74,75,90,91,92]. Then, the relationship between the VD axis (including the VDR and these nutrient metabolism pathways) is quite intriguing. On one hand, VD intake has been generally associated with a decreased risk of developing lung cancer or better survival prognostics. This fact indirectly suggests that the VDR should follow the same pattern when it comes to its presence in the tumoral tissues, having a lower expression at the tumoral site. However, at first glance, the studies seem to have contradictory results regarding this hypothesis [85,87,88,89]. For instance, Kaiser et al. revealed the presence of the VDR in more than half of primary NSCLC tumors, with squamous cell and adenocarcinoma showing the uttermost VDR expression, while another study has shown that the VDR is rather poorly expressed at an mRNA level in lung tumors when compared with normal tissue [93]. In addition, Gheliji et al. [71] found a significant decrease in the VDR presence in tumoral tissues in comparison with adjacent non-cancerous tissues, but only in males, while, interestingly, the same pattern could not be confirmed in females. Also, Li et al. indicated that the Bsm1 (rs1544410 G>A) polymorphism provides significant protection against lung cancer across all genetic models, including allele (OR = 0.62, 95% CI = 0.44–0.87, p = 0.005), homozygous (OR = 0.76, 95% CI = 0.60–0.96, p = 0.019), heterozygous (OR = 0.59, 95% CI = 0.39–0.88, p = 0.010), recessive (OR = 0.80, 95% CI = 0.64–0.99, p = 0.039), and dominant models (OR = 0.57, 95% CI = 0.37–0.86, p = 0.007). Partial protective effects were also found for *Taq1* (rs731236 T>C) and Cdx-2 (rs11568820 T>C) polymorphisms. *Taq1* showed significant protection in the allele (OR = 0.88, 95% CI = 0.79–0.98, p = 0.017) and recessive models (OR = 0.84, 95% CI = 0.73–0.98, p = 0.022). Cdx-2 showed significance in the heterozygous (OR = 0.80, 95% CI = 0.66–0.98, p = 0.032) and dominant models (OR = 0.79, 95% CI = 0.65–0.96, p = 0.018). No significant association was found between the Apa1 (rs7975232 C>A) polymorphism and lung cancer [89]. Considering all these disparities shown in the cited papers above, Menezes et al. [35] proposed a comprehensive study with the aim of assessing VDR expression in its relationship with cancer evolution and subcellular localization. The study concluded that a more developed cancer translates into a predominant nuclear expression of the VDR when compared to the cytoplasmatic levels of this receptor. This may be interpreted in different manners. Firstly, the increased nuclear level may be an indicator of the cell preparations for pathways that regulate tumoral-specific aberrant growth. Secondly, this unbalanced ratio favoring the nuclear presence of the receptor in advanced lung cancers could be a sign of potential or actual activity of VD signaling pathways since the VDR must be transferred to the nucleus for the main VD-cascade reactions to begin and function correctly. Finally, this nuclear expression depreciation, corroborated with elevation of the VDR presence in the cytoplasmic compartment, reveals the presence of active adaptive mechanisms for cells, considering the dramatic changes imposed by a tumoral transition [35].

## 4. Conclusions

Lung cancer remains a significant health challenge worldwide, so understanding its molecular mechanisms facilitates early detection, targeted therapies, and prolonged prognosis.

VD, known for its crucial role in various physiological processes, has emerged as an important inhibitor in carcinogenesis. Mounting evidence has shown that increased levels of VD are associated with a reduced risk of lung cancer, while VD deficiencies are positively correlated with increased susceptibility to the disease, whereas a sufficient intake of VD showed a protective effect against carcinogenesis and inflammation. Additionally, the VDR plays a crucial role in the cellular signaling processes involved in carcinogenesis. Genetic variations in the VDR gene, particularly the polymorphisms Taql, Apal, Bsml, Fokl, and Cdx2, have been associated with an increased risk of lung cancer. It is important to emphasize that certain polymorphisms have shown a stronger correlation with lung cancer across different ethnic groups. However, the relationship between VDR polymorphisms and lung cancer risk remains complex and requires further investigation.

Not only are VD and the VDR particularly important in lung cancer management but also vitamin D-dependent proteins, such as calbindins, calretinin, and S100-G, which are crucial in calcium ion binding and may impact lung cancer. Calbindin 1 suppresses apoptosis in tumor cells, potentially worsening outcomes, but is also linked to better survival and resection success in lung cancer. Calretinin is a mesothelioma marker, complicating lung cancer diagnosis, while CABP9K is overexpressed in various cancers, including lung cancer. Calbindins’ role in inflammation-related diseases suggests they could be significant in lung cancer pathophysiology, offering potential therapeutic targets despite some contradictory findings. These proteins play diverse roles in lung cancer progression and prognosis, making them potential therapeutic targets or diagnostic markers.

This review contributes to the awareness of the role of VD, VD-dependent proteins, and VDR gene variation in lung cancer risk. A comprehensive understanding of these mechanisms is essential for developing personalized prevention and treatment strategies, with further research warranted to overcome the lacunas of the physiopathological mechanisms and to be able to use the findings in clinical practice, ultimately improving outcomes for lung cancer patients. Future clinical applications of vitamin D should focus on its use as an adjuvant therapy to enhance the efficacy of existing lung cancer treatments and as a preventive measure in high-risk populations.

## Figures and Tables

**Figure 1 ijms-25-06664-f001:**
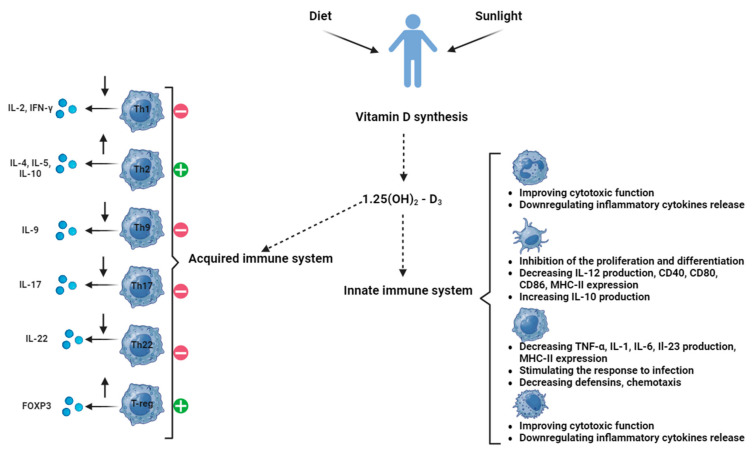
Different roles of vitamin D in immune and cytotoxic processes.

**Figure 2 ijms-25-06664-f002:**
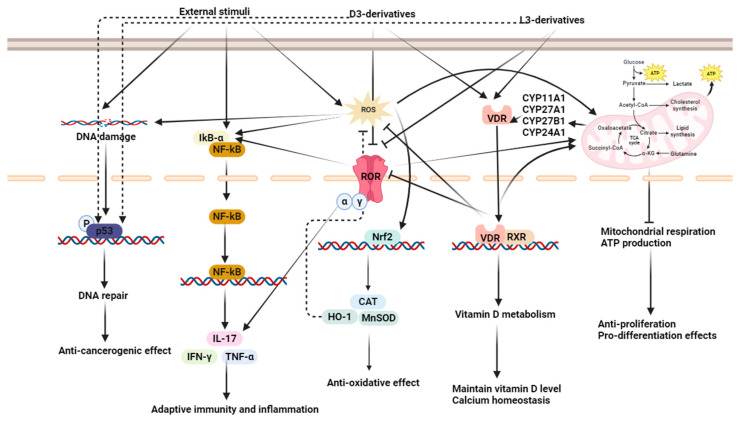
Vitamin D receptor regulatory processes.

**Table 1 ijms-25-06664-t001:** Vitamin D-associated proteomics correlated with lung risk cancer.

Protein	Expression in Tumoral Cells	Lung Cancer Association	References
CALB1	Overexpression	Non-small cell lung cancer	[63]
CALB2	Overexpression	Not associated with cancer but with malignant pleural mesothelioma	[57]
S100-G	Overexpression	Non-small cell lung cancer	[58]

**Table 2 ijms-25-06664-t002:** VDR gene polymorphism correlated with lung risk cancer.

Gene Polymorphism	Genotype/Allele/rs Variant	Ethnicity	Lung Cancer Risk	References
*TaqI*	t allele and TT genotype	Overall population	High risk (especially in Caucasians) t allele was associated with s reduced risk in one study	[74,79]
	rs731236	Asian	High risk	[81]
		African	No correlation	[79]
*Bsml*	B allele, BB and bb genotype	Asian	High risk	[82]
	B allele and bb genotype	Overall population	High risk	[82]
	bb genotype	Caucasian/overall population	High risk	[83]
	rs1544410	Overall population	No correlation, even if a dominant allele may decrease the risk of lung cancer	[81,82]
		Asian	High risk	[82,84]
*ApaI*	Aa and aa genotypes	Overall population	High risk	[81]
	rs7975232	African	High risk	[85]
*FokI*	rs2228570	Overall population	High risk	[86]
		African	High risk	[85]
	f allele	Overall population	Weak correlation with high risk	[87]
	F allele	Overall population	High risk	[88]
*Cdx-2*	rs11568820	Caucasian	Protective factor against lung cancer	[89]
	TT and TT+TC genotypes	Overall population	Low risk	[75]

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
