# Peer review of "Exploring the Role of Vitamin D, Vitamin D-Dependent Proteins, and Vitamin D Receptor Gene Variation in Lung Cancer Risk"

_ijms, 2024, doi:10.3390/ijms25126664_

Round 1

Reviewer 1 Report

Comments and Suggestions for Authors

Requires minor revision.

Comments on the Quality of English Language

Few sentences require rewriting.

Author Response

Dear Reviewer,

Thank you very much for your detailed and constructive feedback. We appreciate the time and effort you put into reviewing our manuscript.

Exploring the Role of Vitamin D, Vitamin D-dependent proteins and Vitamin D

Receptor Gene Variation in Lung Cancer Risk

Overview

The role of VD in lung cancer risk is a topic of ongoing research. There is evidence to

suggest that VD and its dependent proteins may play a role in reducing the risk of lung

cancer, but the exact mechanisms are not yet fully understood. Several studies have found an

association between low levels of VD and an increased risk of lung cancer. Variations in the

VDR gene may also influence an individual's risk of developing lung cancer. Present review

focuses on the highlighting the role of VD, VDR SNPs, and VD associated proteins in lung

cancer.

Major Comments

  1. Line 44 to 50: Authors should include the statistical data of lung cancer associated

with their country rather than stating about only United States.

Thank you very much. The adjustment has been made at line 48-51.

“According to Romanian National Institute of Public Health 2023 report, Romanian men are more prone to be diagnosed with lung cancer, accounting for 17% of the newly reported oncological cases are pulmonary cancer, compared to 7% in women. The same report shows that the number of male smokers is 4 times higher than female smokers (30.6% compared to 7.6%) [1].”

Line 149 to 152: Authors should include the approximate levels of VD for lower and

higher risk of lung cancer. What is the adequate amount of VD required?

The risk associated has been added to line 159-161.

“Zhan et al. defined the daily intake of VD associated with the risk of developing NSCLC: <4µg/d the risk associated with developing NSCLC is the highest, decreasing to more than half after >10 µg/d intake [43].”

  1. Can tabulate the research data highlighting the possible levels of VD categorizing the

low and high risk of lung cancer across various geographies.

Thank you for you comment! Table 2 provides a comprehensive overview of the association between various gene polymorphisms related to Vitamin D and lung cancer risk across different ethnicities.

  1. Line 434: Authors stated that the VD dependent proteins can act as therapeutic targets

or diagnostic markers, but no information or discussion was provided in the review

supporting it.

We have made the adjustments.

“Calbindin 1 suppresses apoptosis in tumor cells, potentially worsening outcomes, but is also linked to better survival and resection success in lung cancer. Calretinin is a mesothelioma marker, complicating lung cancer diagnosis, while CABP9K is overexpressed in various cancers, including lung cancer. Calbindins' role in inflammation-related diseases suggests they could be significant in lung cancer pathophysiology, offering potential therapeutic targets despite some contradictory findings.”

  1. Section 3: Should include more statistical oriented discussion for VDR gene variation

and SNPs reported.

Thank you very much for your feedback. We have included these phrases:

“Also, Li et al. indicate that the Bsm1 (rs1544410 G>A) polymorphism provides significant protection against lung cancer across all genetic models, including allele (OR = 0.62, 95% CI = 0.44-0.87, P = 0.005), homozygous (OR = 0.76, 95% CI = 0.60-0.96, P = 0.019), heterozygous (OR = 0.59, 95% CI = 0.39-0.88, P = 0.010), recessive (OR = 0.80, 95% CI = 0.64-0.99, P = 0.039), and dominant models (OR = 0.57, 95% CI = 0.37-0.86, P = 0.007). Partial protective effects were also found for Taq1 (rs731236 T>C) and Cdx-2 (rs11568820 T>C) polymorphisms. Taq1 showed significant protection in the allele (OR = 0.88, 95% CI = 0.79-0.98, P = 0.017) and recessive models (OR = 0.84, 95% CI = 0.73-0.98, P = 0.022). Cdx-2 showed significance in the heterozygous (OR = 0.80, 95% CI = 0.66-0.98, P = 0.032) and dominant models (OR = 0.79, 95% CI = 0.65-0.96, P = 0.018). No significant association was found for the Apa1 (rs7975232 C>A) polymorphism and lung cancer [99].”

Minor Comments

  1. Line 15: M.C.?

We are sorry, it was a misspelling. The correct name is A.C. (Alexandra Craciun).

  1. Line 24 to 28: Show complete plagiarism.

Thank you for your attention. We made the adjustment and paraphrased it.

  1. Line 53: “air pollution reaching negative milestones annually,” what does “negative

milestones” mean?

We adjusted the phrase.

Furthermore, with air pollution reaching alarming records annually, environmental factors are a significant variable to be taken into consideration.

  1. Line 58: HPV, no need to abbreviate.

Thank you. We removed the abbreviation.

  1. Line 100: “to our organism”? Should rewrite the sentence.

Widely known as a principal regulator of the calcium-phosphate balance, VD is a vital nutrient for many other processes in the human body. Referred to as calciferol, this fat-soluble vitamin is provided both exogenously, from foods like egg yolks, fish, red meat, or liver, and endogenously, during exposure to UVB radiation.

  1. Line 143: “We synthesized the involvement of VD,” Synthesized? What does that

mean in this sentence?

We want to say that : ‘’We summarized the involvement of VD in the interplay between immune and cytotoxic processes in Figure 1.’’

  1. Line 148, 157: “Boughanem et al.”, “Wang et al.”

We corrected.

  1. Line 209: Ca2+

We corrected

  1.  Line 218, 397: NSCLC, no need to abbreviate.

Thank you, we corrected.

Remark

The review is well paced and structured clearly highlighting the role of VD in lung cancer

risk. Minor revision is needed.

Thank you for your implication. The adjustment have been made.

Reviewer 2 Report

Comments and Suggestions for Authors

The paper "Exploring the Role of Vitamin D, Vitamin D-dependent proteins and Vitamin D Receptor Gene Variation in Lung Cancer Risk" presents the roles of vitamin D, vitamin D-dependent proteins and VDR gene variation in lung cancer risk. Genetic changes in the VDR gene, especially Taql, Apal, Bsml, Fokl, and Cdx2 polymorphisms, have been associated with an increased risk of lung cancer. The authors correctly present the mechanisms that may influence the development of lung cancer, which may contribute to the development of personalized strategies for the prevention and treatment of this cancer. Overall, the manuscript is well written. There are no major comments regarding the described scope of knowledge about the biological action of the vitamin D receptor in the context of the immune system and its anti-inflammatory relationship and modulatory effect on carcinogenesis. The literature cited is adequate.

In the conclusions, the authors could add a comment relating to the most promising directions of clinical use of vitamin D in the prevention or treatment of lung cancer.

Author Response

Dear Reviewer,

Thank you for your constructive feedback. Thank you for the time and effort you put into reviewing our manuscript.

The paper "Exploring the Role of Vitamin D, Vitamin D-dependent proteins and Vitamin D Receptor Gene Variation in Lung Cancer Risk" presents the roles of vitamin D, vitamin D-dependent proteins and VDR gene variation in lung cancer risk. Genetic changes in the VDR gene, especially Taql, Apal, Bsml, Fokl, and Cdx2 polymorphisms, have been associated with an increased risk of lung cancer. The authors correctly present the mechanisms that may influence the development of lung cancer, which may contribute to the development of personalized strategies for the prevention and treatment of this cancer. Overall, the manuscript is well written. There are no major comments regarding the described scope of knowledge about the biological action of the vitamin D receptor in the context of the immune system and its anti-inflammatory relationship and modulatory effect on carcinogenesis. The literature cited is adequate.

In the conclusions, the authors could add a comment relating to the most promising directions of clinical use of vitamin D in the prevention or treatment of lung cancer.

Thank you very much for your valuable feedback. We have made the adjustment.

“Future clinical applications of vitamin D should focus on its use as an adjuvant therapy to enhance the efficacy of existing lung cancer treatments and as a preventive measure in high-risk populations.”
